# Physiological and Pathological Functions of Neuronal Hemoglobin: A Key Underappreciated Protein in Parkinson’s Disease

**DOI:** 10.3390/ijms23169088

**Published:** 2022-08-13

**Authors:** Ran Zheng, Yiqun Yan, Jiali Pu, Baorong Zhang

**Affiliations:** Department of Neurology, Second Affiliated Hospital, School of Medicine, Zhejiang University, Hangzhou 310009, China

**Keywords:** neuronal hemoglobin, dopaminergic neuron, mitochondria homeostasis, Hb-α-synuclein complex, iron metabolism, AKT signaling pathway

## Abstract

The expression of Hemoglobin (Hb) is not restricted to erythrocytes but is also present in neurons. Hb is selectively enriched in vulnerable mesencephalic dopaminergic neurons of Parkinson’s disease (PD) instead of resistant neurons. Controversial results of neuronal Hb levels have been reported in postmortem brains of PD patients: although neuronal Hb levels may decline in PD patients, elderly men with higher Hb levels have an increased risk of developing PD. α-synuclein, a key protein involved in PD pathology, interacts directly with Hb protein and forms complexes in erythrocytes and brains of monkeys and humans. These complexes increase in erythrocytes and striatal cytoplasm, while they decrease in striatal mitochondria with aging. Besides, the colocalization of serine 129-phosphorylated (Pser129) α-synuclein and Hb β chains have been found in the brains of PD patients. Several underlying molecular mechanisms involving mitochondrial homeostasis, α-synuclein accumulation, iron metabolism, and hormone-regulated signaling pathways have been investigated to assess the relationship between neuronal Hb and PD development. The formation of fibrils with neuronal Hb in various neurodegenerative diseases may indicate a common fibrillization pathway and a widespread target that could be applied in neurodegeneration therapy.

## 1. Introduction

Parkinson’s disease (PD) is the second most common neurodegenerative disease, which affects more than 6 million people worldwide [1]. The pathological hallmarks of PD are the selective loss of dopaminergic neurons in the substantia nigra (SN) and aggregation of intraneuronal α-synuclein, known as Lewy bodies (LB) and Lewy Neurites (LN) [2]. However, it is difficult to slow down or arrest the progression of PD due to its unclear pathological mechanism [3]. Elucidating novel molecular pathways of pathogenesis and discovering new therapeutic targets for PD are urgent and essential needs. Interestingly, studies have recently shown that hemoglobin (Hb) is expressed and selectively enriched in PD-vulnerable dopaminergic neurons instead of resistant neurons [4]. Moreover, Hb protein directly interacts and colocalizes with α-synuclein in human erythrocytes and brains [4,5,6,7]. Increasing evidence suggests the association between Hb expression and PD development [8,9,10,11,12]. Besides the primary functions of Hb in erythrocytes, neuronal Hb can regulate mitochondria hemostasis, α-synuclein accumulation and iron metabolism in PD pathogenesis [4,13,14]. However, the beneficial or detrimental roles of neuronal Hb in PD are still unclear. In this study, the past and recent studies elucidating the expression, localization, physiological, and pathological roles of neuronal Hb and its relationship with PD progression have been discussed. Furthermore, we present an updated review on how neuronal Hb contribute to neuropathological changes of PD (Figure 1), and propose potentially rational targeted therapies and biomarkers that warrant further investigation.

## 2. Expression of Hemoglobin Chains in Non-Erythroid Cells

Hb is widely known as an iron-containing protein in blood erythrocytes essential for oxygen, carbon dioxide, and nitric oxide transport in mammals [15]. It consists of a tetrameric structure, including two α globin chains (HBA) and two β globin chains (HBB) [16]. Monomeric globin chains contain a heme prosthetic group. Heme is a strong activator of globin chain transcription [17]. Moreover, Hemin, an oxidized form of heme, participates in the regulation of gene transcription, translation of Hb globin molecules, and Hb assembly [18].

Recent studies have shown that expression of Hb α and β chains are found in atypical sites other than blood, including retinal cells [19], epithelial cells [20], sciatic nerves [21], hepatocytes [22], mesangial cells of the kidney [23], and neuronal/glial cells [4]. This indicates that Hb α and β chains are not exclusively restricted to the erythroid lineage, and thus, may play multiple roles in health and disease. For instance, Hb can perform diverse physiological functions in various cells [15]. Hb acts as an antioxidant in mesangial cells of the kidney [23]. Hb and its peptides have antimicrobial activity in vaginal epithelium [20]. Interestingly, separate expression patterns have also been described for the α and β chains in some cell types. For instance, only Hb α chains are expressed in endothelial and peripheral catecholaminergic cells, while only β chains are expressed in macrophages [24,25]. Moreover, Hb β chains can form a tetramer with a higher binding affinity for oxygen than α2β2 heterotetramers [26], suggesting that α and β chains may have independent functions that should be further investigated in detail.

## 3. Existence and Physiological Functions of Hemoglobin in Brain

Recent studies have demonstrated that Hb α and β chains are expressed in neurons of cortex, hippocampus, cerebellum, and retina of rodent brains using robust methods covering cell type-specific transcriptome analyses (cDNA microarrays and nanoCAGE) of laser-captured micro-dissected neurons, quantitative reverse transcriptase-polymerase chain reaction (RT-qPCR), combined in situ hybridization and immunohistochemistry with the neuronal marker neuronal nuclear (NeuN) antigen [4]. Hb α and β chains have also been detected in primary cell cultures of murine ventral midbrain using Western blotting (WB), excluding the possibility of blood contamination [27]. Furthermore, neurons express key regulators of Hb, including erythropoietin (EPO) [28], EPO receptor [29], and hypoxia-inducible factor 1α (HIF1α) [30], suggesting a potential self-regulatory network. Notably, a similar and species-conserved pattern of neuronal Hb expression has been observed in brains of mice, rats, and humans [4,27].

In vivo studies have shown that neuronal Hb α and β chains colocalize in mouse neurons, indicating that intact tetrameric Hb structures exist in the brain [31]. Therefore, neuronal Hb may have partial biological functions similar to those associated with its roles in erythrocytes based on their known roles in blood. For instance, the upregulation of neuronal Hb via injection or transgenic overexpression of EPO enhances brain oxygenation and mitochondrial activity under physiologic or hypoxic conditions, indicating that neuronal Hb is involved in oxygen storage and mitochondrial-associated neuroprotection [32,33]. Moreover, Hb β chains are enriched in cortical pyramidal neurons, which interact with mitochondrial proteins and are involved in mitochondrial energetics, suggesting that Hb β chains have partially similar activities and functions to the intact neuronal Hb [34]. Nevertheless, the physiological functions of neuronal Hb and its peptides are still a matter of debate. Considering the low concentration of neuronal Hb in brain, some scholars have proposed that neuronal Hb is not primarily involved in oxygen storage and supply to mitochondria, but has (pseudo-) enzymatic activity and may be part of signaling pathways in health and disease [35]. However, this needs further clarification.

Additionally, Hb has also been detected in glial cells, including cortical and hippocampal astrocytes, as well as virtually all mature oligodendroglia with much lower expression than in neurons [4,36,37,38]. Hb can regulate oxidative stress and iron metabolism in glial cells [36,39,40]. Besides, the ALGGEN—PROMO online software analysis has indicated the presence of NF-κB binding site on Hb α promoter. Electrophoretic mobility shift assay (EMSA) and Chromatin Immunoprecipitation (ChIP) results have also confirmed that p65-NF-κB enhances the transcription of Hb α chains by binding to Hb α promoter region, suggesting an unrecognized role of Hb in the regulation of glial inflammation [20].

## 4. Unexpected Expression of Hemoglobin in Mesencephalic Dopaminergic Neurons

Mesencephalic dopaminergic (mDA) cells are the major source of dopamine in the brain [41]. The mDA cells have two main groups of projecting cells: A9 cells of the SN and A10 cells of the ventral tegmental area (VTA). The selective degeneration of A9 dopaminergic neurons, involved in the regulation of voluntary movements and postural reflexes, is one of the primary hallmarks of PD [42]. Recent studies identified Hb α and β globin chains in the mDA neurons in the SN and VTA of mouse brain, where Hb is enriched as mRNA and protein in the large majority of A9 neurons, but only present in <5% of A10 neurons [4]. Similar to mouse brain, Hb has also been detected in A9 neurons of human postmortem brain [4,43]. Specifically, both Hb α and β chains have been detected in the soma of neuronal cells, while only β chains are present in dendrites [27]. These observations indicate that neuronal Hb is selectively enriched in PD-vulnerable neurons, suggesting that neuronal Hb may have a role in PD development.

## 5. Subcellular Localization of Neuronal Hemoglobin

It is necessary to determine the location of a molecule in a cell when studying its physiological and pathological functions. As a result, co-immunoprecipitation (Co-IP) and mass spectrometry (MS) technologies have been used to determine the proteins specifically interacting with Hb β chain in neurons. In total, 5 of the 14 Hb β chain binding proteins are mitochondrial, including ATP synthase subunits α and β, mitochondrial malate dehydrogenase (MDH2), ADP/ATP translocase 4, and a mitochondrial phosphate carrier called solute carrier family 25 member 3 (SLC25A3) [34]. Besides, Hb β chain also interacts with other unexpected substances, including histone H3 and lysine-specific histone demethylase 8 (KDM8) [34]. These results indicate that Hb β chains may function in both mitochondria and nucleus, involving mitochondrial energetics and epigenetic regulation, and transmit signals between mitochondria and nucleus.

Furthermore, both Hb α and β chains have been observed to colocalize with mitochondrial markers in the human brain, consistent with the observation that heme is synthesized within the mitochondria [44,45]. Specifically, immuno-electron microscopy (immuno-EM) and WB analysis of proteins prepared from subfractions of human brain-enriched mitochondrial revealed that Hb α and β chains are predominantly localized within the intermembrane space of mitochondria [46]. Hb α and β chains have also been identified in cytoplasm and nucleus of mDA neurons, and maintain in a dynamic equilibrium under physiological condition [4]. However, Hb localization shifts from the intermembrane space to the outer membrane of cerebellar mitochondria in men with a long course of PD, suggesting that dynamic localization of Hb may involve different functional and physiopathological processes [47]. Therefore, further studies should characterize the roles of neuronal Hb in different subcellular localizations to reveal their potential effects on neurodegeneration.

## 6. Alternation of Neuronal Hemoglobin Levels in Aging and PD

Hb levels within the brain decrease with aging [48]. Some scholars have indicated that neuronal Hb levels reflect systemic Hb levels. Systemic Hb levels are decreased in older individuals, possibly due to chronic inflammation or a reduced bone marrow function [49]. Heme has also been detected in neurons and decreases with aging, and thus, may reduce the production of neuronal Hb [50]. Meanwhile, other scholars have hypothesized that the neuronal Hb protein proteolysis increases with aging, leading to the downregulation of Hb levels and accumulation of brain iron [48]. Interestingly, thermal stability of Hb increases with the aging of red blood cells (RBCs), indicating that Hb conformation is altered during the aging process [51]. However, this remains to be confirmed in neurons.

Similar to aging, decreased Hb levels have been detected in the blood of PD patients, which positively correlates with the severity of PD and iron metabolism [52]. Meanwhile, decreased neuronal Hb levels have also been reported in postmortem brains of PD patients [53]. Double-labeling immunofluorescence and confocal microscopy have revealed that neuronal Hb α and β chains are reduced in about 80% of the α-synuclein-deposited dopaminergic neurons in SN and vulnerable neurons of the medulla oblongata containing abnormal α-synuclein-immunoreactive small inclusions or Lewy pathology (LP) in postmortem brains of PD patients [53]. Notably, this loss of neuronal Hb in pathologically deposited neurons of PD is specific, as neuroglobin and EPO receptors have been found to be equally present in neurons with and without abnormal protein inclusions [53]. Furthermore, neuronal Hb protein levels have been found to decrease in mitochondria of the striatum of monkey and human brains in an age-dependent manner. This decrease is negatively correlated with the intracellular α-synuclein accumulation and Hb-α-synuclein complex formation [6,7].

However, although neuronal Hb levels may decline in PD patients, elderly men with higher Hb levels have an increased risk of developing PD [8]. This observation may support the hypothesis that increased Hb degradation promotes PD pathogenesis, as evidenced by increased brain iron in PD brains [54]. Besides, an emerging study indicated that levels of Hb chains, including α, β, γ1, and γ2, possibly related to defects in iron, are elevated in brains of PD patients based on glycomics and proteomics [12]. Therefore, given the controversial reports of neuronal Hb levels in brains of PD patients, further investigations assessing neuronal Hb levels in prodromal and different stages of PD are needed to elucidate the role of neuronal Hb in PD.

## 7. Direct Interaction between α-Synuclein and Neuronal Hemoglobin

α-synuclein is abundantly expressed in the central nervous system and peripheral tissues [55,56]. Interestingly, over 99% of blood α-synuclein is found in RBCs [57]. Some studies have demonstrated that oligomers of α-synuclein are toxic by themselves [58]. However, other studies have indicated that α-synuclein pathogenicity is related to the interactions with other protein partners [59,60]. Notably, Co-IP has detected a direct interaction between α-synuclein and Hb protein, which form a complex in both brain striatum and RBCs of monkeys and humans [6,7]. The levels of Hb-α-synuclein complex decrease in striatum mitochondria, but increase in cytoplasm with aging. However, these complex levels are increased in both mitochondria and plasma when cultured dopaminergic cells are treated with α-synuclein [6,7], indicating that neuronal Hb and α-synuclein have distinct regulatory patterns during aging and PD. Furthermore, a biotinylation by antibody recognition (BAR) technique has been used to label LP for determining protein interactions and subcellular localization of insoluble cellular components [61]. BAR-Serine 129-phosphorylated (Pser129) α-synuclein identifies enriched protein HBB localized to LP in synucleinopathy samples when BAR-SYN1 (total α-synuclein) in brain tissues of synucleinopathies and non-synucleinopathies have similar protein abundance patterns [5], which further confirms the interaction between α-synuclein and neuronal Hb and suggests a potential influence of Hb in α-synuclein pathology.

However, the binding mechanisms are unclear. In vitro experiments have shown that oxidative stress, which has been detected in the brain and RBCs of PD patients, promotes the binding between α-synuclein and Hb through the formation of dityrosine crosslinks [62]. It is difficult to determine molecular interactions principally arising from the insoluble nature of LP. Therefore, further studies using cryo-electron microscopy (cryo-EM) and MS are required to explore the specific binding sites between α-synuclein and Hb and its binding mechanisms.

## 8. Potential Mechanisms Linking Hemoglobin to PD Development

Several studies have assessed whether Hb alteration in PD is only a pathological phenomenon or a potential driver/regulator of PD development (Figure 2).

### 8.1. Neuronal Hemoglobin and Mitochondrial Homeostasis

The selective vulnerability of dopaminergic A9 neurons in PD is in part due to the high metabolic and energetic demands caused by a massive, highly branched axonal arbor with hundreds of thousands of transmitter release sites [63,64]. This is particularly relevant to the primary physiological functions of Hb, which involve oxidative stress regulation and mitochondrial homeostasis [65]. For instance, gene expressions that modify oxygen homeostasis and mitochondrial oxidative phosphorylation are changed when mouse dopaminergic cell line MN9D is transfected with Hb α and β chains. Specifically, 46% of the upregulated genes encode subunits of mitochondrial complex I–V [4]. Mitochondrial complex I plays a central role in PD, and deficits in its subunits and activities have been consistently detected in the SN of PD patients [66]. Furthermore, mitochondrial toxin rotenone (a toxin used to induce PD in animal models) significantly decreases mRNA expression of Hb chains in dopaminergic neurons of rats, accompanying the inhibition of mitochondrial complex I [27]. As a result, the accumulation of free oxygen radicals and mitochondrial dysfunction occur, suggesting that Hb is neuroprotective in regulating the expression of mitochondrial complex I and mitochondrial homeostasis in PD patients.

Neuronal Hb levels in mitochondria and cytoplasm are normally maintained in a dynamic equilibrium [7]. However, WB results from mitochondrial and cytoplasmic fractions have indicated that mitochondrial/cytoplasmic ratios of Hb α and β chains are significantly decreased in PD brains compared with controls. In contrast, immunohistochemistry studies have shown that the quantity of Hb-containing mitochondria is increased in SN neurons that survive to late stages in PD brains [46], suggesting that free mitochondrial Hb is increasingly converted to insoluble Hb that is ignored by WB assays in neurons of PD patients. Moreover, accumulation of α-synuclein in the cytoplasm impedes the translocation of Hb from the cytoplasm to mitochondria, resulting in lower levels of free functional mitochondrial Hb [7]. Mitochondrial Hb can increase mitochondrial membrane potential (MMP), which plays a key role in ATP production [67]. While exogenous α-synuclein reduces neuron MMP, Hb overexpression enhances free Hb levels in mitochondria, stabilizes MMP, and lowers α-synuclein-induced neuron apoptosis, indicating that free mitochondrial Hb has a protective role in mitochondrial homeostasis and neuron death [7]. However, considering the increased risk of PD in older men with abnormally high Hb levels and selective vulnerability of dopaminergic A9 neurons with enriched neuronal Hb [4,8], neuronal Hb may participate in a complex and dynamic regulatory network by changing its expression, solubility, and subcellular localization during PD development, rather than a merely mitochondrial protective role.

### 8.2. Neuronal Hemoglobin and α-Synuclein Accumulation

Misfolding and aggregation of α-synuclein, resulting in the formation of filamentous cellular inclusions in dopaminergic neurons, is a hallmark of PD [68]. Stereotactic injection of adeno-associated virus (AAV) carrying Hb α and β chains into the substantia nigra pars compacta (SNc) of mouse brain induces the formation of endogenous α-synuclein C-terminal truncated species, which accumulates in pathological inclusions, and promotes α-synuclein aggregation and toxicity. An in vitro study has detected similar α-synuclein fragments in iMN9D cells overexpressing Hb α and β chains after treatment with preformed α-synuclein fibrils, indicating that overexpression of neuronal Hb promotes α-synuclein pathogenesis [14]. Furthermore, long-term Hb overexpression in SNc induces the loss of about 50% of dopaminergic neurons, a mild motor impairment, and deficits in recognition and spatial working memory, suggesting that neuronal Hb plays a critical role in PD progression [14].

Besides truncating α-synuclein and promoting α-synuclein pathology, neuronal Hb accumulates in the nucleolus of iMN9D cells as insoluble aggregates after treatment with neurochemical 1-methyl-4-phenylpyridinium (MPP+) or rotenone. An in vivo experiment has also detected similar Hb aggregates in SNc of mice brain, where Hb is overexpressed after AAV transfection [13]. Moreover, neurochemical stress-induced epigenetic modifications of DA neurons, nucleolar stress, and autophagy impairment are dependent on Hb expression [13].

Importantly, multiple studies have shown that proteins capable of self-aggregating into macromolecules can promote the accumulation of disease-causing proteins [69,70]. Increased macromolecular crowding has effects on protein structure, promoting protein aggregation in a time- and concentration-dependent way [71]. Significant conformational changes and Hb protein aggregates formation have been observed upon incubation with the crowding agent bovine serum albumin (BSA) in a concentration-dependent manner [71]. This suggests that α-synuclein accumulation facilitated by Hb overexpression may in turn promote Hb aggregation by increasing macromolecular crowding, thus inducing a vicious cycle of protein aggregation and promoting PD pathology.

Additionally, accumulation of α-synuclein in the cytoplasm increases the neuronal Hb-α-synuclein complex formation in cytoplasmic and mitochondrial fractions [7]. However, whether this complex contributes to PD pathology is unclear. The specific regulation networks of neuronal Hb aggregation, α-synuclein accumulation, and Hb-α-synuclein complex formation in PD requires further investigations.

### 8.3. Neuronal Hemoglobin and Iron Metabolism

Disrupted iron metabolism promotes PD pathogenesis [72]. Hb is the most abundant source of iron in humans [73]. A transcriptomic meta-analysis of blood microarrays has revealed that genes related to Hb and iron metabolism are significantly downregulated in PD patients [9]. Blood Hb protein and iron levels are also decreased in PD patients, and Hb expression levels are significantly correlated with iron metabolism [52]. Although numerous studies have suggested that decreased serum/plasma iron levels are associated with the risk of PD, a non-significant trend towards higher cerebrospinal fluid (CSF) iron levels have been detected in PD patients [74], suggesting an overall iron misdistribution between central and peripheral regions in disease progression. For instance, total iron and iron (III) levels increase in the SN of PD patients by 176% and 225% compared with age-matched controls, respectively [75]. Unbound excess cellular iron species in the brain can be cytotoxic, promoting oxidative stress [76], mitochondrial protein dysfunction [77], α-syn aggregation [78], neuronal cell death, and lipid peroxidation (involved in a recently discovered ferroptosis process) [79,80,81].

However, the relationship between neuronal Hb levels and iron accumulation in the brain has not been fully assessed. A recent study reported that Hb overexpression in dopamine cell lines alters gene transcription related to iron metabolism, including ferritin heavy chain 1 (FTH1) and transferrin receptor (TFRC) [4]; thus, this may contribute to iron accumulation in SN. Furthermore, haptoglobin (Hp), a Hb-binding protein that determines free Hb levels based on different phenotypes [82], has common genetic polymorphisms, including Hp 1-1, Hp 2-1, and Hp 2-2. Among them, Hp 2-1 phenotype is associated with higher PD risk [83]. PD patients with Hp 2-1 phenotype have low serum iron [82], suggesting that Hp phenotypes may be associated with Hb and iron level abnormalities and reflect neuronal Hb and iron regulation in PD patients.

Notably, males and females have different methods for managing iron in bodies, and thus, this should be taken into consideration when analyzing iron regulation [84]. For instance, in males, there is no active mechanism for removing iron from the body, which makes them more likely to be diagnosed with PD [47,84,85]. There is also a difference between males and females when it comes to the sub-mitochondrial localization of neuronal Hb. Females have a more normal distribution of mitochondrial Hb in comparison to males [47]. These gender differences suggest a potential regulatory relationship between neuronal Hb localization and iron metabolism in PD progression, which needs to be investigated further.

### 8.4. Neuronal Hemoglobin and GH/IGF-I Signaling Pathway

Growth hormone (GH) and insulin-like growth factor-1 (IGF-1) are both growth factors playing trophic roles in the case of neuronal regeneration [86]. GH secretion decreases with advancing age, and impaired GH release leads to severe structural and functional changes in the brain [87,88,89]. According to a study of hormone plasma profiles, untreated idiopathic PD patients had significantly lower GH levels than healthy age-matched controls, suggesting a contributory role for GH in disease progression [90]. GH acts on tissues primarily mediated by GH receptors and the downstream secretion of IGF-I [91]. The levels of IGF-I also decrease in an age-dependent manner; however, examinations of the clinical correlations in PD cohorts have indicated that serum IGF-I levels are elevated at PD onset, suggesting an ongoing compensatory or “fight-to-injury” mechanism [92]. Basic and translational data have shown that GH/IGF-I signaling pathway has protective and homeostasis roles in PD [93,94]. Nonetheless, the relationship between this signaling pathway with disease onset, duration, and severity requires further investigations, and studies on the mechanisms underlying the effects of GH and IGF-1 on PD are still limited.

Notably, GH and IGF-I are known to significantly affect circulating and neuronal Hb levels, indicating a potential regulatory relationship between GH/IGF-I axis and neuronal Hb in PD [95]. In central nervous system (CNS), neurons respond strongly to GH by activating highly expressed GH receptors [91]. In response to GH receptor activation and dimerization, Janus kinase 2 (JAK2) is activated, followed by phosphorylation of signal transducer and activator of transcription 5 (STAT5) [96,97]. GH also induces the secretion of IGF-I, whose activation is mediated by phosphatidylinositol 3-kinase (PI3K)/protein kinase B (AKT) signaling pathway [98]. Interestingly, experiments have revealed that JAK2/STAT5 activation is involved in EPO-induced Hb synthesis [99]. In addition, interaction of the IGF-I receptor with the AKT pathway phosphorylates major Hb transcriptional regulator GATA-binding factor 1 (GATA-1), which is specifically expressed in A9 neurons and directly regulates the levels of Hb and α-synuclein [100,101]. α-synuclein, in turn, has been reported to interact with AKT and enhance the solubility and plasma localization of AKT in response to IGF-1. Eliminated, mutant, and hyper-expressed α-synuclein evokes the dysregulation of the AKT signaling cascade, strongly indicating that the interplay of IGF-I and α-synuclein on AKT-GATA-1-regulation has implications for Hb dysregulation and PD development [102]. The PI3K/AKT signaling pathway can also induce HIF1α expression, thus regulating Hb expression and maintaining the balance between oxygen demand and supply [103,104]. Furthermore, a substantial decrease in neuronal HBB transcript has been observed in GH-deficient rats compared with intact rats, while GH or IGF-I administration robustly upregulated the levels of HBB transcripts in rat brain [95,105]. Given the impaired of GH/IGF-I secretion in PD and the strong correlation between GH/IGF-I axis activation and neuronal Hb expression, it is possible that GH/IGF-1 axis-induced Hb regulation is involved in PD development. However, it is still necessary to determine how this GH/IGF-I signaling pathway regulates Hb and α-synuclein expression, which may provide novel target for PD therapy.

### 8.5. Hemoglobin and Vitamin D

As a fat-soluble secosteroid, vitamin D exerts its effects by binding to the vitamin D receptor to regulate cellular growth, immunity, metabolism, and erythropoiesis [106,107,108]. Several studies have demonstrated that serum vitamin D levels were inversely associated with the risk and severity of PD [109,110]. Following the discovery that the SN is highly expressed of vitamin D receptor and the enzyme that converts vitamin D to its active form, it has been speculated that insufficient circulating vitamin D levels in the SN may lead to dysfunction or cell death [111,112]. Animal studies indicated that vitamin D restored dopaminergic circuits by promoting the release of glial cell-derived neurotrophic factor (GDNF) in PD [113,114]. In addition, a cell culture study demonstrated that calcipotriol, a potent vitamin D analogue, suppressed calcium-dependent α-synuclein aggregation by inducing calbindin-D28k expression [115]. However, the pathophysiologic role of vitamin D in PD is not yet fully understood.

Notably, recent studies have reported correlations between vitamin D levels and major hematological parameters, which reflect blood flow and are important for tissue oxygenation [116,117,118]. Interestingly, there are conflicting data on hematological parameters regarding Vitamin D replacement. While Arabi et al. demonstrated that vitamin D supplementation had no significant effect on hemoglobin concentration [119], Doudin et al. concluded that 25(OH) Vitamin D levels were inversely correlated with hemoglobin concentration and mean corpuscular hemoglobin levels [120]. Considering the strong correlation between Vitamin D and hemoglobin, the potential effect of Vitamin D on neuronal hemoglobin needs further exploration to better understand the issue of vitamin D in PD.

### 8.6. Hemoglobin and α-Synuclein in Erythrocyte

Although whether serum and plasma levels of total α-synuclein are increased or decreased in PD is controversial [121,122,123], total α-synuclein levels are markedly positively correlated with Hb levels in plasma. Inclusion of Hb levels as covariates in the analysis of total plasma α-synuclein improves the ability of immunoassays to detect a difference between PD patients and controls [124]. Moreover, the complex of Hb and α-synuclein has been detected in human erythrocytes and brains, which increases in an age-dependent manner [6,7]. However, no study has evaluated the amounts and relationships of erythrocytic and neuronal complex of Hb and α-synuclein in PD patients.

α-synuclein in erythrocytes is approximately 1000-fold higher than in cerebrospinal fluid, and thus, can be a potential source of pathogenic α-synuclein in brain [125,126]. Erythrocyte-derived extracellular vesicles (EVs) containing α-synuclein can penetrate the blood–brain barrier (BBB) and induce microglial inflammatory responses [127]. Notably, erythrocytic Pser129 α-synuclein levels are significantly increased in PD patients than in healthy controls. These Elevated levels of Pser129 α-synuclein have been found to be positively correlated with disease duration, Hoehn & Yahr stage (H&Y), and UPDRS III score [128]. Additionally, patients with PD also have higher plasma levels of oligomeric α-synuclein and a panel of posttranslational modified forms of α-synuclein [129]. Importantly, the plasma EVs from PD patients have increased levels of oligomeric α-synuclein than those from controls, suggesting a peripheral-to-central pathogenesis for PD [130]. However, a further investigation is required to determine whether the complex of Hb and α-synuclein in erythrocytes of patients with PD can be secreted as EVs and contribute to α-synuclein pathology in the CNS.

## 9. Erythropoietin/Neuronal Hemoglobin Axis as a Potential Target for PD Therapy

Existing therapies for PD are only symptomatic as they do not provide permanent or disease-altering effects [131]. Numerous studies have demonstrated that EPO counteracts many of the processes altered in PD, including neuroinflammation, oxidative stress, mitochondrial dysfunction, and cell death [132,133]. In rotenone, 6-hydroxydopamine or 1-methyl-4-phenyl-1,2,3,6-tetrahydropyridine (MPTP)-induced rodent models of PD, EPO treatment restored the levels of tyrosine hydroxylase (TH) through the modulation of neuroinflammation [134,135,136]. In addition, a recent study reported that EPO prevented MPP+-induced alterations in mitochondrial morphology and compensated for the loss of mitochondrial complex I activity [137]. Although several in vitro studies of PD have indicated that EPO improved neuronal cells viability by modulating autophagy or apoptosis-related pathways [138,139], the mechanisms by which EPO affect neurons are complex and largely unknown.

In the CNS, EPO may interact with at least four distinct isoforms of its receptor to exert different functions. Among them, the canonical isoform, which is primarily found in the hematopoietic system, is also present in the brain and regulates hemoglobin synthesis [140,141]. Importantly, a specific N-terminally truncated isoform of the EPO receptor has been found in A9 DA neurons of the SN [142]. Although the mechanism of this truncated isoform-induced action has not been characterized, it would be interesting to assess whether the DA-EPO receptor is involved in neuronal Hb regulation, given that Hb is selectively highly expressed in A9 DA neurons. This may allow the development of novel isoform-selective drugs targeting the EPO/neuronal Hb axis, leading to a more specific therapy for PD.

## 10. Commonality of Amyloid Fibrils with Hemoglobin in Neurodegenerative Diseases

Pathological deposits of filamentous protein in neurons and glia have been detected in multiple neurodegenerative diseases, including α-synuclein in PD and amyloid-β (Aβ) and tau in Alzheimer’s disease (AD) [69]. These neurodegeneration studies have identified “one conformer per disease” paradigm, with the presence of unknown buried cofactors and diverse patterns of posttranslational modification mediating the structural diversity of fibrillar polymorphs [143,144]. Notably, Hb chains have been detected in various neurons and glial cells in the brain and are associated with different proteinopathies of neurodegenerative diseases [145,146]. For instance, besides the formation of Hb-α-synuclein complex in PD brains, in vivo interactions between Hb and Aβ or hyperphosphorylated tau have also been detected in the brains of mice and humans with AD [147]. However, although Hb overexpression promotes the accumulation of α-synuclein and impairs DA neurons, EPO treatment increases the Hb expression and reduces Aβ42-induced cognitive decline and tau hyperphosphorylation in mice [148,149]. Furthermore, while an in vitro study has found that Hb promotes Aβ oligomer formation [150], oligomeric soluble Aβ species-induced proinflammatory activation of astrocytes reduces when Aβ is associated with Hb [151], suggesting that neuronal Hb and Hb-contained fibrils have complex and conflicting roles in neurons and glias in neurodegenerative diseases. Currently, it is unclear whether the affinity between the Hb chains and α-synuclein in PD is beneficial or detrimental and whether pathogenicity arises from gain or loss of function. It is also unclear whether the formation of amyloid fibrils containing Hb represents a primary pathogenic process or a non-specific secondary phenomenon in different neurodegenerative diseases.

## 11. Conclusions and Future Prospects

In summary, Hb chains are selectively enriched in A9 neurons instead of A10 neurons, and are associated with brain aging and PD risk. Potential mechanisms of neuronal Hb regulation related to mitochondrial homeostasis, α-synuclein accumulation, iron metabolism, and GH/IGF-I signaling pathway in PD have been proposed. Although overexpression of neuronal Hb may be neuroprotective by maintaining mitochondrial homeostasis, it may also promote α-synuclein aggregation, iron deposition, and dopaminergic neuron death. Colocalization and binding of Hb and α-synuclein occurs in erythrocytes and brains of PD. Commonality of amyloid fibrils with Hb is found in various neurodegenerative diseases. However, it is still unclear whether neuronal Hb and Hb-α-synuclein complex are beneficial or detrimental in PD development.

Although Hb is significantly associated with PD development, its role in PD has not been appreciated yet. Therefore, based on published findings on Hb, studies should be performed to elucidate the roles of Hb:In non-erythrocytes, Hb α and β chains levels are not comparable. It is important to study the interplay of various transcription factors involved in regulating Hb chain expressions, depending on their specific functions in different non-erythrocytes. Hb or its constituents may have different functional significance, wherein only HBA or HBB, but not the entire Hb tetramer, are required, which should be confirmed.Given the conflicting roles of neuronal Hb overexpressions in PD development, evaluation of soluble and insoluble neuronal Hb protein expressions in PD neurons may indicate functional and dysfunctional forms of Hb, as well as elucidate the effects of loss or gain of functions of Hb. Moreover, the alterations of neuronal Hb expression and function at different disease stages should be clarified.Expressions of Hb can be found in neuronal mitochondria, nucleus and the surrounding cytoplasm. Changes in the structure of Hb and balance of its dynamic localization may be involved in PD pathogenesis. Under physiological and pathological conditions, clarification of the role of Hb in different subcellular localizations may reveal the mechanisms of Hb displacement and transfer.Hb interacts directly with α-synuclein in vivo. However, the cross-seeding ability of Hb in α-synuclein aggregation has not been evaluated in vitro. Although neuronal Hb overexpressions induced α-synuclein truncation and increased its toxicity, the effects of neuronal Hb overexpressions in PD progression have not been investigated in vivo. It is yet to be established whether neuronal Hb acts as a contributory factor in PD pathogenesis by increasing Pser129-α-synuclein expressions and promoting LP formation.Given that Aβ-associated proinflammatory activations of astrocytes were reduced when Aβ bound to Hb, the significance of the Hb-α-synuclein complex in glial cell activation and its overall impact in disease progression should be confirmed.Formation of Hb-α-synuclein complexes in both erythrocyte and neuron cytoplasm as well as their similar quantitative trends with aging raise the question of whether the alterations in their structure and interactions are consistent in peripheral and central tissues of PD patients. It would be helpful to investigate the correlation between peripheral Hb-α-synuclein complex burden and disease progression, which may inform on novel non-invasive peripheral biomarkers that reflect pathological alterations in the brain and reveal disease progression stages when a patient is still alive.The IGF-I/AKT signaling pathway is involved in regulating Hb as well as α-synuclein expressions and is also regulated by different forms of α-synuclein in turn. Elucidation of the mechanisms of this regulatory network in PD neurons may reveal novel therapeutic targets.Several co-interactive proteins have been reported in fibrils of different neurodegenerative diseases. The significance of these cofactors, either as contributory factors in disease pathogenesis or if they only represent a consequence, should be evaluated. Given the formation of fibrils composed of neuronal Hb in diverse neurodegenerative diseases and the involvement of neuronal Hb in disease progression, it is possible to identify a common fibrillization pathway and a widespread structural target via advanced cryo-EM techniques, which have allowed scientists to determine the structures of amyloid fibrils at near-atomic resolution [152,153].

## Figures and Tables

**Figure 1 ijms-23-09088-f001:**
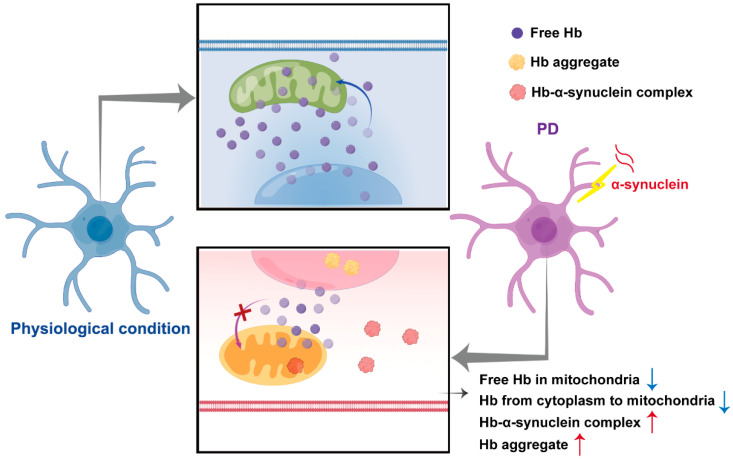
Hypothetical schematic presentation of neuronal hemoglobin regulatory networks in physiological and Parkinson’s disease conditions. In the physiological condition, neuronal Hb was widely localized in cell bodies, including the mitochondria, nucleus, and the surrounding cytoplasm. However, when dopaminergic neurons were challenged with α-synuclein, formation of Hb-α-synuclein complexes was induced in the cytoplasm and mitochondria, accompanied by reduced translocations of Hb from the cytoplasm to the mitochondria, reduced levels of free mitochondrial Hb and aggregation of Hb in the nucleus, which may contribute to neuronal death and pathological progression of PD.

**Figure 2 ijms-23-09088-f002:**
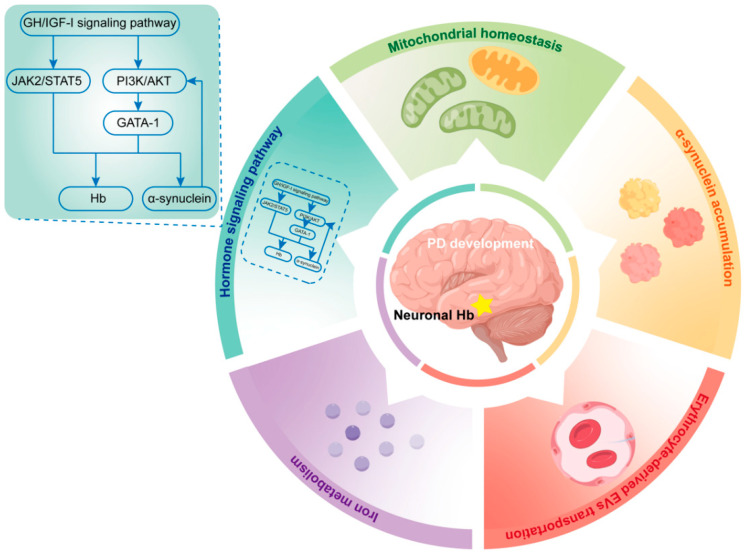
Potential mechanisms through which hemoglobin is involved in PD development. Several potential mechanisms were associated with the roles of hemoglobin to PD development. These mechanisms were established to be involved mitochondrial homeostasis, α-synuclein accumulation, iron metabolism, hormone regulation, and erythrocyte-derived EVs transportation. Hb, hemoglobin; PD, Parkinson’s disease; GH, Growth hormone; IGF-I, insulin-like growth factor-I; JAK2, Janus kinase 2; STAT5, signal transducer and activator of transcription 5; PI3K, phosphatidylinositol 3-kinase; AKT, protein kinase B; GATA-1, GATA-binding factor 1.

## Data Availability

Not applicable.

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
