# Peer review of "Physiological and Pathological Functions of Neuronal Hemoglobin: A Key Underappreciated Protein in Parkinson’s Disease"

_ijms, 2022, doi:10.3390/ijms23169088_

Round 1
Reviewer 1 Report
This review describes an association between mitochondrial dysfunction and hemoglobin and chronic neurodegeneration. It is potentially interesting. The authors may additionally discuss the impact of vitamin D and its impact of blood perfusion parameters due to its steroid like chemical structure in the context of this review. At the current stage this review only focus on the interaction of various well known causes and mechanisms of chronic neurodegeneration without reflections on possible influencing therapeutic options.
Author Response
Dear reviewer 1,
We gratefully appreciate for your critical comments on our manuscript. Based on these comments, we have made careful modifications on the original manuscript and provided responses to each of the points.
Please see the attachment.
Yours sincerely,
Baorong Zhang

Reviewer 2 Report
It is a very interesting review in which Zheng et al., recapitulate how neuronal hemoglobin contribute to neuropathological changes of Parkinson Disease. It is well organized, written and easy to follow, the references are up to date and it is of high interest to IJMS readers.
To make it easier for readers to understand the problem, I think that the authors should present in the introduction, in a general way, the neuropathological changes present in Parkinson's disease in order to later address the problem of HB. And on the other hand in point 8.4 authors should explain if the GH/IGF-I axis is really relevant in the Hb and PD relationship.
Author Response
Dear reviewer 2,
We gratefully appreciate for your thoughtful suggestions on our manuscript. Based on these suggestions, we have made careful modifications on the original manuscript and provided responses to each of the points.
Please see the attachment.
Yours sincerely,
Baorong Zhang

Round 2
Reviewer 1 Report
accpet now